# Floquet control of interactions and edge states in a programmable quantum simulator

Or Katz [1,3,5] ✉, Lei Feng[1,4,5] ✉, Diego Porras[2] & Christopher Monroe [1]

Quantum simulators based on trapped ions enable the study of spin systems and models with rich dynamical phenomena. The Su-Schrieffer-Heeger (SSH) model for fermions in one dimension is a canonical example that can support a topological insulator phase when couplings between sites are dimerized, featuring long-lived edge states. Here, we experimentally implement a spin-based variant of the SSH model using one-dimensional trapped-ion chains with tunable interaction range, realized in crystals containing up to 22 interacting spins. Using an array of individually focused laser beams, we apply site-specific, time-dependent Floquet fields to induce controlled bond dimerization. Under conditions that preserve inversion symmetry, we observe edge-state dynamics consistent with SSH-like behavior. We study the propagation and localization of spin excitations, as well as the evolution of highly excited configurations across different interaction regimes. These results demonstrate how precision Floquet engineering enables the exploration of complex spin models and dynamics, laying the groundwork for future preparation and characterization of topological and exotic phases of matter.

Topological quantum materials have attracted extensive interest across physics and materials science[1], offering potential applications as robust information carriers of quantum information[2]. Among these, topological insulators serve as prime examples of such materials and give rise to macroscopic properties that qualitatively differ between the material's bulk and its edges, impacting phenomena such as quantum coherence of excitations and transport[3,4]. Perhaps the simplest instance of a topological insulators is the Su-Schrieffer-Heeger (SSH) model in one dimension[5–7], describing a crystal of fermions with alternating bond strengths. Under conditions preserving chiral symmetry, this model can support long-lived edge states and host a topologically non-trivial ground state.

Quantum simulators based on photonic and neutral atomic systems, have enabled the realization of SSH-like models and the observation of edge-state dynamics[8–13]. These systems typically rely on engineered hopping terms or synthetic dimensions, but often lack tunable long-range interactions or full site-resolved control. In this paper, we implement a variant of the SSH model for a spin system[14] using a trapped-ion quantum simulator, offering programmable interaction range and site-resolved preparation and measurement of spin states. This enables the exploration of edge phenomena and spin dynamics in previously inaccessible regimes.

Trapped-ion systems enable the bottom-up construction of quantum materials in one or more dimensions, using strong electromagnetic confinement and laser-cooling of individual atoms[15–17]. Laser-driven optical dipole forces couple internal spin states to collective modes of motion, generating tunable spin-spin interactions that can be long- or short-ranged, and either uniform or staggered in sign[18,19]. A global optical dipole force has been used to simulate a variety of quantum spin states and phases, including ferromagnetic,

[1]Duke Quantum Center, Department of Physics and Electrical and Computer Engineering, Duke University, Durham, NC, USA. [2]Institute of Fundamental Physics IFF-CSIC, Madrid, Spain. [3]Present address: School of Applied and Engineering Physics, Cornell University, Ithaca, NY, USA. [4]Present address: Department of Physics, Fudan University, Shanghai, China. [5]These authors contributed equally: Or Katz, Lei Feng. ✉e-mail: or.katz@cornell.edu; leifeng@fudan.edu.cn

antiferromagnetic, disordered "XY", and continuous symmetry breaking[18,20,21]. While this platform supports universal Hamiltonian engineering (and universal quantum computation)[18,22,23], previous ion trap simulators have typically employed optical dipole force patterns that lack the spatial symmetry and structure necessary to realize models akin to the spin-based SSH model[14], which requires a high-resolution, alternating bond configuration across the chain.

Floquet engineering with optical fields, or the periodic optical driving of a material, enables dynamic control over microscopic interaction terms and the shaping of band structures in a wide range of systems[24–28]. Floquet Topological Insulators, for example, are systems that acquire their topological properties solely through periodic driving. However, realizing such states in solid-state systems is challenging due to the small lattice spacing and the high intensities required. In atomic platforms, Floquet fields have been used to engineer time crystals[29,30] and dynamical gauge fields[31]. However, prior implementations have typically employed global driving, which limits both the complexity and the spatial resolution of the resulting model.

Here, we use a programmable trapped-ion quantum simulator, with individual optical control over each spin to implement a site-dependent Floquet drive and explore SSH-like dynamics[14]. Using an array of tightly focused laser beams that simultaneously address each individual atom in the crystal, we independently control the amplitude and phase of local Floquet fields across the chain, enabling tunable bond dimerization with interaction patterns that preserve inversion symmetry. We observe dynamics consistent with edge-state localization in one-dimensional chain containing up to 22 interacting spins, contrasting with thermalization behavior in the bulk. By varying the interaction range and preparing spin configurations with multiple excitations we explore dynamics beyond the single excitation subspace. The ability to initialize the system with site-specific spin excitations, probe their evolution, and program the spatial structure of the effective interactions via Floquet control offers a versatile platform for exploring interacting spin models, with potential extensions toward probing topological signatures and exotic quantum phases.

## Results

### Long-range SSH spin model

The spin-based SSH model[14] studied here takes the form of a long-range XY spin Hamiltonian with a coupling matrix $\mathcal{J}_{ij}$ between spins $i$ and $j$.

$$H = \sum_{i,j} \mathcal{J}_{ij} \left( \hat{s}_+^{(i)} \hat{s}_-^{(j)} + \hat{s}_-^{(i)} \hat{s}_+^{(j)} \right), \tag{1}$$

where $\hat{s}^{(j)}$ are spin-$\frac{1}{2}$ operators acting on site $j$. The coupling matrix $\mathcal{J}_{ij}$ exhibits a dimerized structure, in which the interaction strengths alternate in magnitude across the chain. We define a two-sublattice structure: odd-numbered sites belong to sub-lattice A and even-numbered sites to sub-lattice B. Interactions between spins on opposite (or same) sublattices follow a dimerized pattern, forming an alternating bond structure between even and odd bonds (see Supplementary Fig. 1 for illustration).

The original SSH model for fermions[5–7] includes only nearest-neighbor hopping terms, and its topological properties and the presence of zero-energy edge states depend on the relative strength of the alternating bonds. The modified SSH model we consider generalizes this concept to spin systems with long-range interactions, provided that the extended dimerization pattern preserves inversion and time-reversal symmetries as formally defined in the "Methods". It has been shown that this spin model can support a topologically nontrivial ground state under specific configurations, characterized by a nonzero Zak phase[14]. Moreover, the model predicts the emergence of edge-localized excitations with longer lifetimes than their bulk counterparts.

The long-range nature of the spin model is particularly interesting because, in its fermionic representation, it gives rise not only to long-range hopping terms but also to interaction terms beyond the quadratic level. These go beyond the standard fermionic SSH model and become especially relevant in configurations with multiple excitations. This model thus provides a platform to study how both the interaction range and the presence of higher-order terms influence dynamics.

### Experimental apparatus

We implement the spin-based SSH model using a trapped ion quantum simulator. We use chains of $^{171}$Yb$^+$ ions, with either $L = 12$ or $L = 22$ spins, confined within a linear Paul trap on a chip[32–34]. Each ion possesses an effective spin created from two "clock" levels within its electronic ground-state $(|\uparrow_z\rangle \equiv |F=1, M=0\rangle$ and $|\downarrow_z\rangle \equiv |F=0, M=0\rangle)$[35]. Our approach involves the use of a equally-spaced array of tightly focused laser beams, in conjunction with an orthogonal global beam. This combination enables the simultaneous driving of Raman transitions between the spin states of individual ions. The Raman addressing method is attuned to the motion along the wavevector difference between the individual and global addressing Raman beams and generates a spin-spin interaction Hamiltonian mediated via the collective motional modes of the ion chain. We initialize and measure the spins using optical pumping and state-dependent fluorescence techniques[35], and the collective motional modes of the ion chain are initialized to near their ground state through sideband cooling[36].

The spin-spin interaction we implement, in the presence of a large transverse field, takes the form of an effective long-range XY Hamiltonian, with interaction bond matrix $J_{ij}$ between spins $i$ and $j$ whose range is controlled[18]. To achieve the bond structure for the SSH model, we first render the spin bonds between nearest-neighboring spins nearly uniform, by controlling the local laser amplitudes at each ion, correcting for the non-uniform participation of ions in the driven phonon modes (see "Methods"). We then control the dimerization between the spins by introducing periodic Floquet fields that manifest as site-dependent transverse magnetic fields $B_z^{(j)}(t)$ oscillating at frequency $\omega$ and with a local site-dependent phase $\varphi_j$ and scaled amplitude $\bar{\eta}$ (see "Methods"). These Floquet fields modify the spin-spin Hamiltonian and in the high-frequency limit $\omega \gg |J_{ij}|$ suppress bonds between spins driven by unequal local field amplitudes between sites $i$, $j$ where $\varphi_i \neq \varphi_j$. We apply a periodic phase pattern $\varphi_j = \frac{\pi}{2}j + \phi$ for $1 \leq j \leq L$, effectively making the transformation $J_{ij} \to \mathcal{J}_{ij}$ (see Eq. (7)). This structure renders the Hamiltonian in Eq. (1) inversion symmetric, and supports the formation of edge state for $\phi = \frac{3\pi}{4}$[14]. In Fig. 1a–c, we display the measured bond matrices for a $L = 12$ crystal for different amplitudes of the Floquet drive; see "Methods" for details of the bond-strength measurement. Increasing the Floquet drive from $\bar{\eta} = 0$ to $\bar{\eta} = 1$ effectively suppresses the odd bonds in comparison to the even ones, allowing for control over the dimerization of bonds in the chain.

### Dynamics of a single-spin excitation

We first examine the effects of Floquet dressing on a solitary spin excitation positioned at the edge of the crystal. In Fig. 1d–f, we present the measured evolution dynamics of each spin $\langle \hat{s}_z^{(j)}(t) \rangle$ governed by the Hamiltonian in Eq. (1) for a crystal comprising $L = 12$ spins, all oriented downwards, except for the spin at edge $j = 1$ that is aligned upwards along the spin $z$-axis. The temporal progression is normalized with respect to the mean nearest-neighbor bond strength $J$ in the absence of the Floquet drive. In the absence of the Floquet field ($\bar{\eta} = 0$), the spin excitation tends to thermalize with the chain over time, as illustrated in Fig. 1d. With an increase in the strength of the Floquet field ($\bar{\eta} = 0.6$, Fig. 1e), the process of thermalization noticeably decelerates. When the chain is fully dimerized with $\bar{\eta} = 1$, the spin excitation remains confined to the edge and ceases to thermalize, as exemplified in Fig. 1f. We present the edge spin's magnetization quantitatively as a function of the drive's amplitude in Fig. 1g. Additionally, we present the excitation

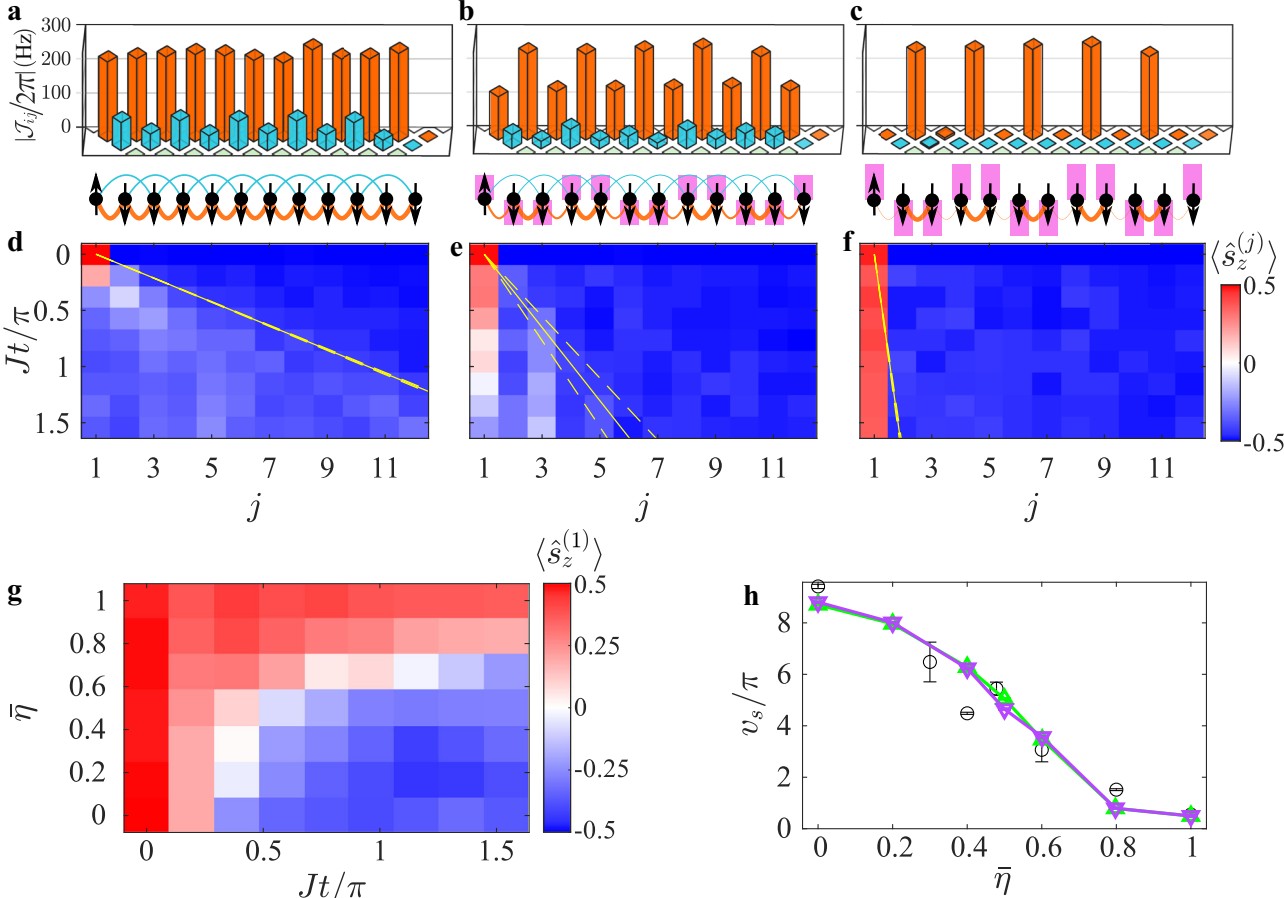

**Fig. 1 | Floquet engineering of spin bonds and topological edge-states. a–c** Bond dimerization in an $L = 12$ spin crystal. Measured spin-spin bond strength $\hbar|\mathcal{J}_{ij}|$ between $i$th and $j$th spins up to next-nearest neighbors (NNN) with increasing Floquet amplitude $\bar{\eta}$. **a** No modulation ($\bar{\eta} = 0$). **b** Moderate modulation ($\bar{\eta} = 0.6$). **c** Full modulation ($\bar{\eta} = 1$). Black spheres for spins and lines for bonds strength, pink rectangles for local Floquet fields. **d–f** Evolution of a single spin excitation at the edge over time for configurations (**a–c**) respectively. The Floquet field suppresses thermalization into the bulk, leading to edge localization. $J$: average nearest-neighbor spin bond coupling absent the Floquet drive; yellow lines show excitation spreading rate (see "Methods"). **g** spin excitation at the crystal edge $\langle \hat{s}_z^{(1)} \rangle$ with varying Floquet field amplitude. **h** Excitation spreading rate $v_s$ (black circles) decreases with increasing Floquet field amplitude, consistent with the modified SSH model (Eq. (1), purple) and the numerical time-dependent ion Hamiltonian (Eq. (2), green). Bars represent $1\sigma$ binomial uncertainties. Data in **d–f** aligns with both numerical models in Supplementary Fig. 2.

spreading rate in Fig. 1h (black), which is extracted from to the slope of the yellow solid lines shown for instance in Fig. 1d–f (see "Methods"). The measured values align closely with two theoretical models, both computed numerically via unitary evolution of the spins either with the SSH spin-Hamiltonian (Eq. (1)) or with the time-dependent spin Hamiltonian the ions experience (Eq. (2)). The results of these models are shown in Supplementary Fig. 2 and in Fig. 1h (purple and green curves). Deviations between the experimental data and the models in Fig. 1h are likely caused by drifts in the absolute value of $J$, which affect the observed excitation propagation speed. Further details regarding the theoretical model can be found in the "Methods" section.

We now turn to compare the response of excitations at the edges with those in the crystal's bulk. We repeated our experiment using a strongly dimerized crystal ($\bar{\eta} = 0.8$) with $L = 22$ spins and a longer interaction range; see "Methods" for experimental details. We considered all possible configurations of single-spin excitations at sites $1 \leq j \leq 22$ as shown in Supplementary Fig. 3 and Fig. 2a ($j = 1$) and Fig. 2b ($j = 11$). Our findings reveal that while bulk excitations rapidly thermalize, edge excitations feature greater isolation. To quantify this difference, we calculated the late time-averaged magnetization of each excited spin at site $j$, denoted as $\bar{s}_{z,j} = 2 \int_{1.5}^{2} \langle \hat{s}_z^{(j)}(\tau) \rangle d\tau$, with $\tau = Jt/\pi$, as depicted in Fig. 2c (black circles). In comparison, the gray dashed line denotes the measured late-time-averaged excitation of the crystal, $\bar{s}_z = \frac{1}{L-1} \sum_{i \neq j} \bar{s}_{z,i}$,

computed by averaging over all crystal sites except the initially excited site $j$. The observed dynamics indicate enhanced persistence of spin excitations at the edges ($j = 1$ or $j = 22$) compared to the bulk ($2 \leq j \leq 21$).

The observed protection of spins near the edges depends on the bond dimerization pattern induced by the Floquet fields. To investigate this, we probe the magnetization of the initially excited spin as a function of the global phase of the Floquet fields $\phi$, while keeping the modulation amplitude fixed at $\bar{\eta} = 0.8$ for the $L = 22$ crystal. In Fig. 2d, we examine the case of a single edge excitation ($j = 1$), and find maximal protection at $\phi = \frac{3\pi}{4}$. This observation is consistent with the theoretical predictions, which show that at this value of the Floquet phase $\phi$, the Hamiltonian becomes inversion symmetric, the Zak phase is quantized and nonzero, and the system is expected to host a nontrivial topological phase in its ground state[14].

In contrast, the evolution of a spin initialized in the bulk (with $j = 11$), as shown in Fig. 2e, is largely insensitive to $\phi$, and rapidly loses its excitation for all values of the Floquet phase. These results highlight the importance of the site-dependent dimerization pattern induced by the Floquet field to enhancing the lifetime of edge excitations.

## Dynamics of multiple spin excitations

Our ability to manipulate the spin states and the range of interactions offers opportunities for exploring dynamics beyond single-spin

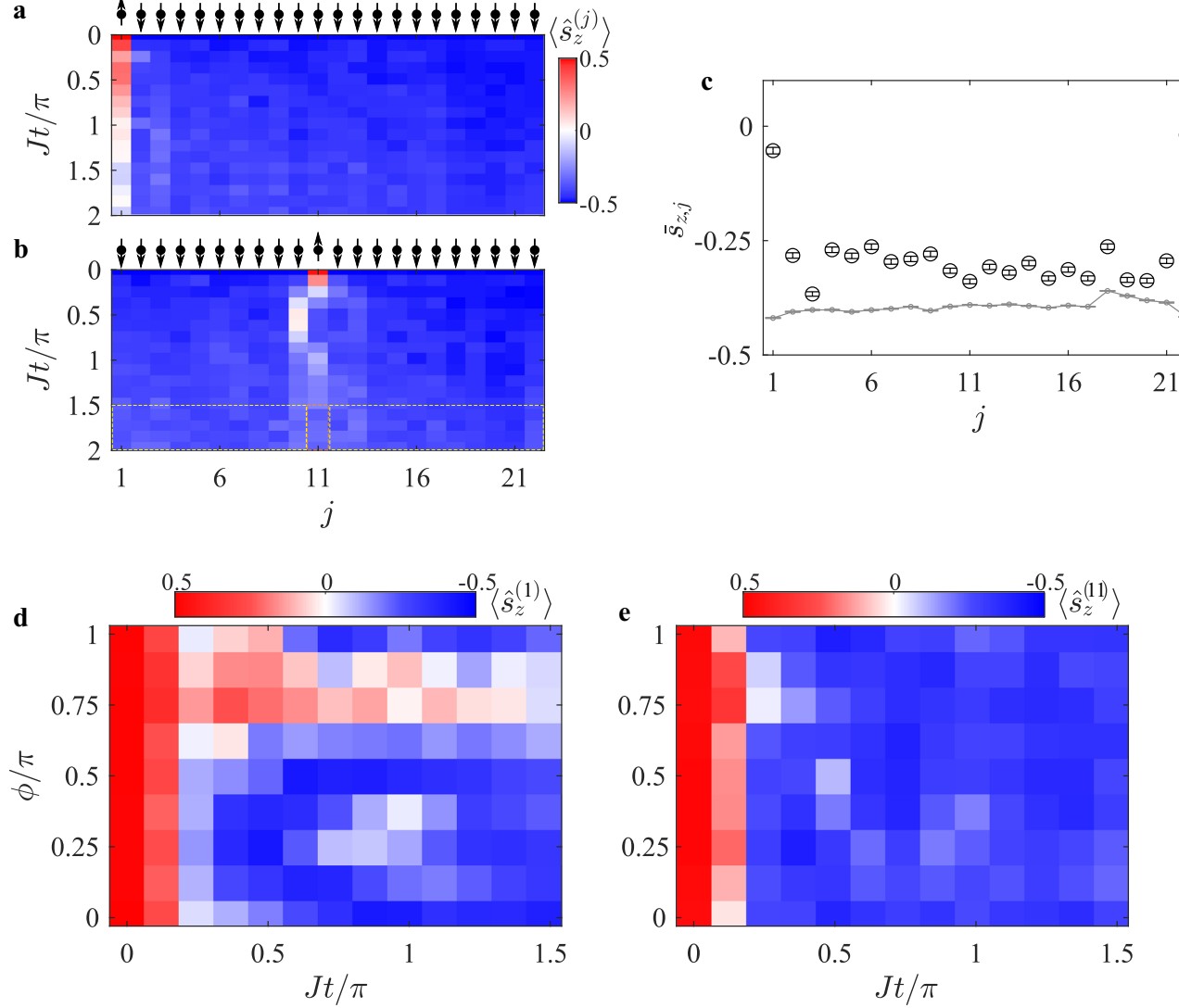

**Fig. 2 | Edge states in a $L = 22$ spin crystal.** Evolution of a single spin-excitation in a strongly-dimerized ($\bar{\eta} = 0.8$) crystal of $L = 22$ spins. **a** Edge-excitation ($j = 1$) and **b** bulk-excitation ($j = 11$). We repeat these experiments for all $1 \leq j \leq 22$ single-spin excitations, and for each calculate the late-time-averaged spin $\bar{s}_{z,j}$ excitation by averaging the values in the orange rectangle; see Supplementary Fig. 3. **c** Late-time-averaged $\bar{s}_{z,j}$ (black circles) highlights the protection of edge excitations over the bulk. Gray line shows the mean late-time excitation across all crystal sites; each point is calculated by averaging the measurement values in the yellow boxes,

excluding the orange box, as exemplified for $j = 11$ in (**b**). Bars represent $1\sigma$ binomial uncertainties. Evolution of edge excitation ($j = 1$, **d**) and bulk excitation ($j = 11$, **e**) as a function of the global Floquet phase shift $\phi$. **d** The spin excitation at the edge survives longer at $\phi = \frac{3\pi}{4}$. The generalized SSH model in ref. [14] predicts a nontrivial topological phase at this value. While we do not directly measure a topological invariant, our observations are consistent with edge-state behavior in such a phase. **e** The bulk excitation is minimally-affected by the Floquet phase.

excitation. In Fig. 3a, b, we depict the evolution of an $L = 12$ spin crystal initialized in a Néel state, where spins at odd (even) lattice sites point upwards (downwards). Here we set the spin-spin interaction range to be long-range (see "Methods"). While absent the Floquet fields ($\bar{\eta} = 0$) all spin excitations thermalize rapidly (Fig. 3a), in the presence of Floquet fields ($\bar{\eta} = 0.6$ in Fig. 3b and $\bar{\eta} = 1$ in Fig. 3c) the thermalization of excitations greatly slows down, predominantly near the edges of the crystal.

To interpret the multi-excitation dynamics and assess the role of fermionic interactions, we compare the experimental results with two models. The first is the SSH spin Hamiltonian defined in Eq. (1), which, when mapped to a fermionic representation, includes interaction terms beyond quadratic order (see Eqs. (14) and (15) in the "Methods" section). The second model describes free fermionic evolution with long-range hopping but excludes beyond-quadratic interaction terms (see "Methods"). We find qualitatively distinct behavior between the two models, particularly for the bulk spins, as exemplified in Fig. 3d–f

and shown comprehensively in Supplementary Fig. 4. This agreement underscores the importance of interaction terms in shaping the dynamics: they enhance thermalization and diminish the robustness of edge excitations, though they do not fully eliminate edge-state protection. To further highlight these findings, we replicate these measurements and analyses for a crystal with short-range interactions, as shown in Supplementary Fig. 5. In this case, we observe good qualitative agreement with the free fermionic model, consistent with the expectations of the standard SSH model—a free fermionic model.

Before concluding, we present another intriguing multiple-excitation scenario where the spin excitations are ordered to form two domains, separated by a single domain wall. The left half of the $L = 12$ crystal are oriented upwards and those in the right half point downwards. As we increase the amplitude of the Floquet drive ($\bar{\eta} = 0$ in a, $\bar{\eta} = 1$ in b), for a short-range interacting crystal, we observe the suppression of thermalization of the two domain walls and the exchange of excitations between the spins at the boundary becomes

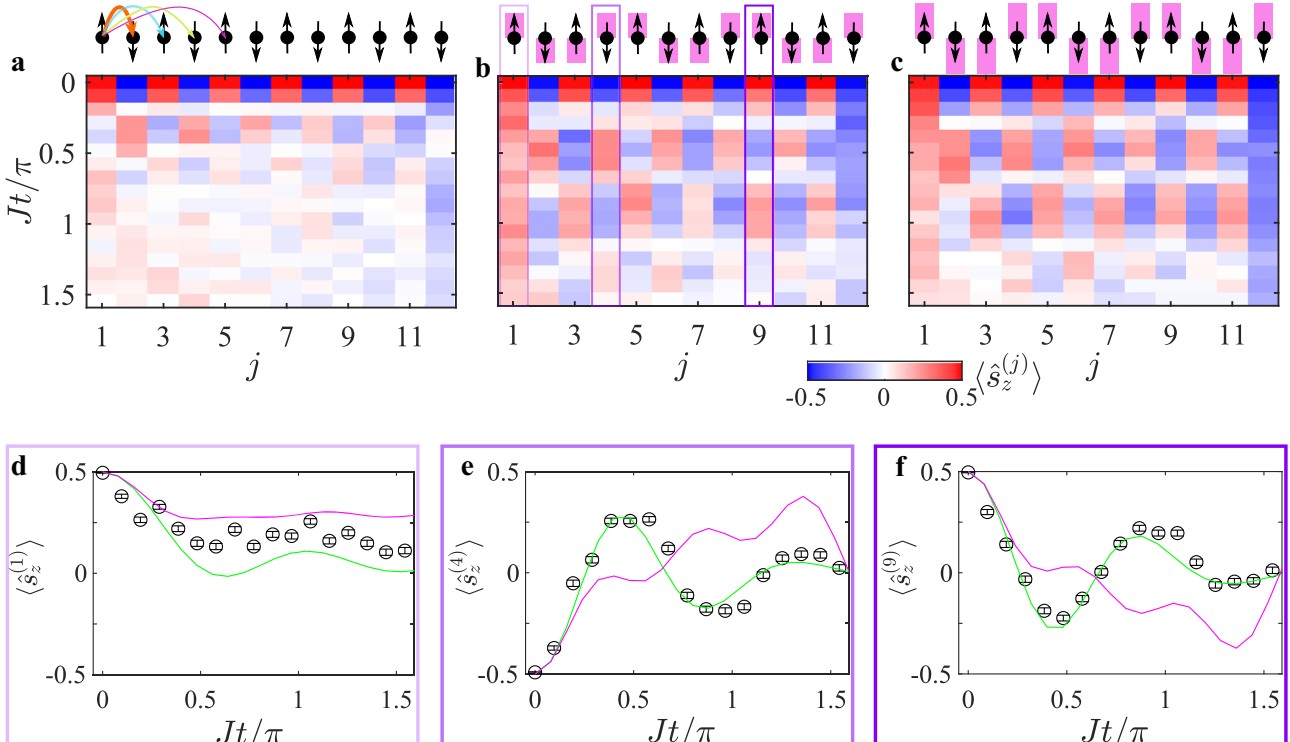

**Fig. 3 | Signatures of fermionic interaction terms.** Evolution of a $L = 12$ spin crystal with long-range interactions, initialized in a staggered spin state. **a** In the absence of Floquet fields ($\bar{\eta} = 0$), the spin excitations quickly hop and thermalize. With full Floquet modulation (**b** $\bar{\eta} = 0.6$ and **c** $\bar{\eta} = 1$), thermalization is partially suppressed and the edges become more protected. Simulation of the $\bar{\eta} = 0.6$ configuration (subplot **b**) of spins $j = 1$ (**d**), $j = 4$ (**e**) and $j = 9$ (**f**). Experimental data (black circle with error bars) agree well with a full simulation of an equivalent fermionic Hamiltonian containing interaction terms (green curve). Bars represent $1\sigma$ binomial uncertainties. For the bulk spins, the data does not agree with a long-range free-fermionic Hamiltonian (magenta curve). The dynamics of all spins are shown in Supplementary Fig. 1. These results are contrasted with short range spin models which are fully explained by free-fermionic evolution. See text and Supplementary Fig. 5.

evident, as shown in Fig. 4. This result qualitatively agrees with the SSH model, which assumes nearest-neighbor interactions. Performing the same experiment for a bond matrix with a significantly longer interaction range ($\bar{\eta} = 0$ in c, $\bar{\eta} = 1$ in d) yields qualitatively different results. We observe a combination of thermalization and boundary excitation, attributed to the bonding of distant spins. These experimental results align well with our numerical spin model, as shown in Supplementary Fig. 6 and also with the spin-based SSH model (see Supplementary Fig. 7). The dynamics of the domain walls are explained by the coherent oscillation between the initial state and the edge-states at the boundary of each of the two domains, as the interaction matrix element between those states dominates over the coupling the bulk (see "Methods"). Our experimental results thus demonstrate, for the first time, the creation of edge states around a boundary set by the initial state rather than system parameters.

## Discussion

In summary, we study the dynamics of a trapped-ion spin chain by applying site-resolved optical Floquet fields that control spin-spin interactions with high precision and generate programmable bond dimerization. Our experiments demonstrate thermalization of spin excitations and reveal that localized edge excitations exhibit dynamics distinct from those in the bulk. By preparing multi-excitation states and tuning the interaction range, we explore how many-body effects and long-range couplings influence the evolution, highlighting the potential for probing complex quantum dynamics in engineered spin models.

This work opens new avenues for engineering spin-spin interaction graphs in low-dimensional quantum systems. In particular, the

ability to suppress selected bonds via Floquet control, independently of the phonon mode structure or the underlying Ising interaction matrix, enhances the controllability of spin-spin coupling patterns. This technique could enable the realization of interaction graphs with quasi-crystalline order[37,38], or the simulation of effective time-dependent magnetic fields that approximate Heisenberg interactions[39] with the aid of additional longitudinal field terms[40]. These models host intriguing quantum phases that have not yet been observed experimentally and may now become accessible using our approach.

Beyond interaction control, our system uniquely combines long-range interactions with site-resolved Floquet fields, enabling future exploration of topological properties in out-of-equilibrium regimes. Such experiments could extend this work by preparing topological ground states or measuring the Zak phase, as proposed in ref. 14. The ability to temporally modulate bond strengths during evolution offers a promising route to study the interplay between topology, interactions, and non-equilibrium dynamics.

Our approach relies on independent frequency control of each Raman beam in the array. We anticipate that future upgrades to the optical setup, including expanding the number of individually addressable beams and allowing variable beam spacing, will enable access to larger systems sizes and more complex spin models.

## Methods
### Additional experimental details
We implement the long-range XY Hamiltonian by applying the Ising spin-spin interaction Hamiltonian $H_{XX}$ in the presence of a large transverse magnetic field Hamiltonian $H_Z$, corresponding to

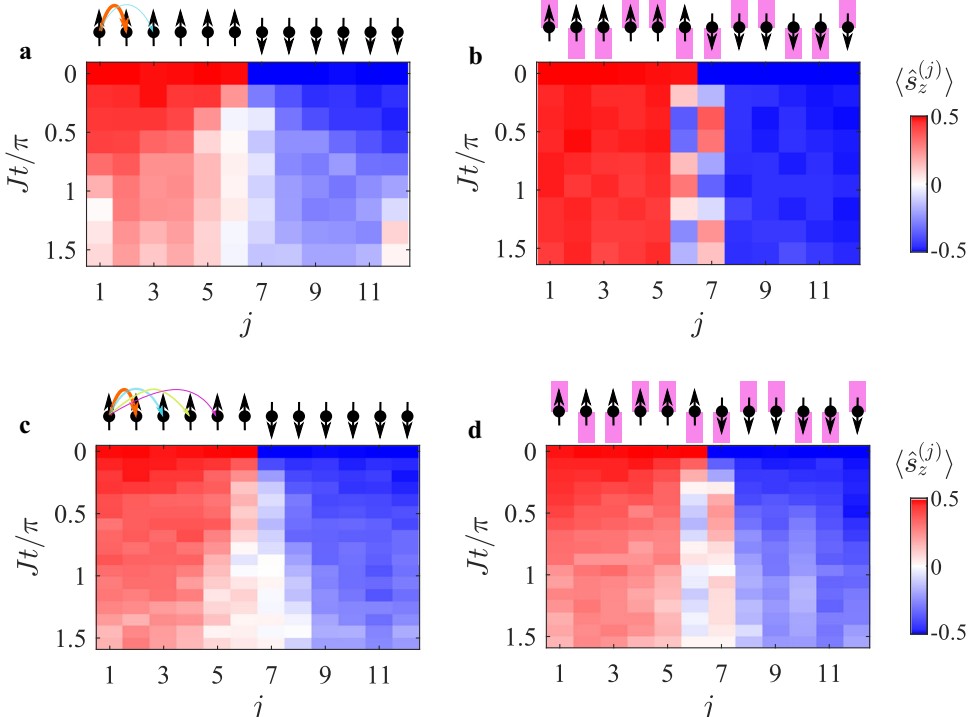

**Fig. 4 | Domain wall dynamics.** The evolution of a domain wall in a $L = 12$ spin crystal, comprising of multiple spin-excitations. **a**, **b** Short range spin-spin interaction, and **c**, **d** Long-range spin-spin interaction. **a**, **c** Absent the Floquet field ($\bar{\eta} = 0$), the domain wall thermalizes as excitations are free to hop. **b**, **d** Fully dimerized crystal ($\bar{\eta} = 1$). **b** Boundary excitations between the wall are exchanged coherently, and the domain wall maintains its state in (**b**) or partially thermalizes in (**d**). The results in **a**–**d** agree well with numerical calculation of the spin evolution, see Supplementary Fig. 6.

the time-dependent Hamiltonian:

$$H(t) = H_{XX} + H_Z(t). \qquad (2)$$

The Ising term is generated via Raman transitions mediated by pairs of beams, which virtually excite collective motion of the ions. One optical beam passes through an acousto-optical modulator (AOM). The AOM is simultaneously driven by two radio-frequency (RF) signals, which split the optical beam into two components with distinct frequencies. These components are then imaged onto the ion chain to address all ions globally. Another beam is split into an array of more than thirty tightly focused beams, passing through a multi-channel AOM. Twenty-two independent RF signals control the frequency and amplitude of 22 beams, with all other beams being blocked. This combination concurrently drive the first red and blue sideband transitions in the dispersive regime. When the detunings from the radial collective motional modes are symmetric, this configuration realizes the Ising Hamiltonian:

$$H_{XX} = 2 \sum_{i,j}^{L} J_{ij} \hat{s}_x^{(i)} \hat{s}_x^{(j)} \qquad (3)$$

with a symmetric spin-spin interaction matrix $J_{ij}$[18].

The $J_{ij}$ coupling elements we realize are real-valued, and can be calculated using the expression[18,21]:

$$J_{ij} = \sum_{k=1}^{N} \frac{\eta_{ik} \eta_{jk} \Omega_i \Omega_j}{2(\Delta + \omega_N - \omega_k)}. \qquad (4)$$

Here, $\eta_{ik} = 0.08 b_{ik}$ are the Lamb-Dicke parameters describing the coupling between spin $i$ and motional mode $k$ through the Raman transitions, with $b_{ik}$ as the mode participation matrix elements[34]. $\Omega_j$ are positive real-valued parameters representing the resonant carrier Rabi

frequency at ion $1 \le j \le L$ for each applied tone, and $\omega_k$ represents the motional frequencies along one radial direction, labeled in decreasing order with $1 \le k \le N$. The $L = 12$ ($L = 22$) spin crystal is constructed using the middle ions in a chain of $N = 15$ ($N = 27$) atoms, with $N - L$ auxiliary ions located near the edges of the chain. For the $N = 15$ chain, one auxiliary ion is positioned at the left end of the $L = 12$ crystal spin (site $j = 0$), and two ions are placed at the right end (sites $j = 13, 14$). In the case of the $N = 27$ chain, there are two auxiliary ions on the left side (sites $j = -1, 0$) of the $L = 22$ spin crystal, along with an additional three ions on the right edge (sites $j = 23, 24, 25$). These auxiliary ions participate in collective motion via their Coulomb coupling to the other ions, but are not illuminated by the Raman beams (i.e., their equivalent $\Omega_j$ is set to zero). Consequently, the evolution is independent of the auxiliary ions' spins, as their $J_{ij}$ matrix elements are identically zero, and they are not considered as part of the spin crystal described in the main text. Their presence contribute to increasing the trapping potential near the edges of the crystal, ensuring nearly uniform spacing between the $L$ inner ions, with an average distance of approximately $3.75\,\mu m$. This uniformity is crucial for aligning with the uniformly-spaced array of individually addressing beams. We note that, in principle, the number of auxiliary ions can be odd or even. Had the experiment included the capability to dynamically adjust the beam positions to accommodate non-uniform ion spacings, the use of auxiliary ions could have been avoided. Incorporating such a capability in future setups could provide an alternative approach.

The trap frequencies and mode participation factors are determined by the trapping potential. We employ a quadratic trapping potential in the radial direction with center-of-mass frequency of $\omega_1 = 2\pi \times 3.08\,MHz$ and an axial electrostatic potential of $V(x) = c_4 x^4 + c_2 x^2$. For the 15-ion chain, the coefficients are $c_2 = 0.11\,eV/mm^2$ and $c_4 = 1.6 \times 10^3\,eV/mm^4$, while for the 27-ion chain,

they are $c_2 = -0.1$ eV/mm² and $c_4 = 235$ eV/mm⁴. Here, $x$ is the coordinate along the chain axis. These potentials determine the frequencies of the collective modes of motion. The effective wave-vector of the optical field is aligned to selectively drive only one specific set of radial modes. For the $N = 15$ ion chain, the frequencies are $\omega_k \in \{3.08, 3.07, 3.05, 3.03, 3.01, 2.98, 2.96, 2.93, 2.90, 2.88, 2.85, 2.83, 2.80, 2.78, 2.78 | 1 \le k \le 15\} \times 2\pi$ MHz. In the case of the $N = 27$ ion chain, the frequencies of the collective motional modes are as follows: $\omega_k \in \{3.08, 3.07, 3.07, 3.06, 3.05, 3.04, 3.03, 3.02, 3.00, 2.99, 2.98, 2.96, 2.95, 2.93, 2.92, 2.90, 2.89, 2.87, 2.86, 2.84, 2.83, 2.82, 2.81, 2.80, 2.79, 2.77, 2.77 | 1 \le k \le 27\} \times 2\pi$ MHz.

We control the range of interaction of the realized $J_{ij}$ matrices by adjusting the Raman beat-note detuning $\Delta$, which is measured relative to the transition of the least-frequency radial (zig-zag) motional mode. We further achieve relatively uniform spin bond strengths (in the absence of the Floquet field) by controlling the amplitudes of individual beams, compensating for the rapid spatial variation of the participation matrix elements of low- frequency phonon modes.

In this work, we implemented three different interaction matrices. The first interaction matrix $J_{ij}$, calculated using Eq. (4) for the $L = 12$ spin crystal ($N = 15$ ion chain) is depicted in Supplementary Fig. 8a. It was achieved with $\Delta = -99 \times 2\pi$ kHz, $J = 0.25 \times 2\pi$ kHz and by normalizing the beams' amplitudes in the array to follow $\Omega_j/\Omega_1 = \{1.0, 1.0, 0.65, 0.87, 0.69, 0.97, 0.74, 0.97, 0.68, 0.86, 0.65, 0.99 | 1 \le j \le 12\}$. This configuration results in a short-range interaction matrix, which decays approximately exponentially and is staggered in its sign, as expected from detuning near the zig-zag mode[19]. We estimated the average bond strength between spins $i$ and $j$ as $\bar{J}(d) = \frac{1}{L-d} \sum_{n=1}^{L-d} |J_{n,n+d}|$ where $d = |i-j|$ represents the distance between the two spins. The calculated $\bar{J}(d)$ is presented in Supplementary Fig. 8d (diamonds) and fitted with a curve $\bar{J}(d) = 3.9J \times e^{-1.36|i-j|}$. This experimental configuration was used in generating Figs. 1, 4a, b and Supplementary Figs. 2, 5g–l, 6g–l. To correct for the alternating bond sign pattern in $J_{ij}$ and make the sign uniform, we apply a staggered optical phase shift to the individual beam array, similar to ref. 41. Specifically, we shift the phase of all odd-site beams by $\pi$ (using the multi-channel AOM), which is mathematically equivalent to transforming the transverse spin operators as $\hat{s}_x^{(i)} \to -\hat{s}_x^{(i)}$ and $\hat{s}_y^{(i)} \to -\hat{s}_y^{(i)}$ for odd $i$. Since this transformation leaves the longitudinal spin operator $\hat{s}_z^{(i)}$ unchanged, it does not affect the applied transverse field.

The second interaction matrix $J_{ij}$ for the $L = 12$ spin crystal ($N = 15$ ion chain) is illustrated in Supplementary Fig. 8b. It exhibits a longer-range interaction, with all coefficients having the same positive sign. This matrix is realized by detuning the Raman beatnote near the center of mass (COM) mode, with $\Delta = 29 \times 2\pi$ kHz $+ \omega_1 - \omega_N$ and $J = 0.25 \times 2\pi$ kHz. We use a relative Rabi frequency profile $\Omega_j/\Omega_1 = \{1.0, 1.0, 1.10, 1.07, 1.16, 1.10, 1.17, 1.10, 1.16, 1.07, 1.10, 1.0 | 1 \le j \le 12\}$ to make the nearest neighbor (NN) bonds nearly uniform in strength. The calculated $\bar{J}(d)$ is presented in Supplementary Fig. 8d (asterisks) and approximated by the curve $\bar{J}(d) = 1.5J \times e^{-0.42|i-j|}$. This configuration was used to generate Fig. 3 and Supplementary Figs. 5a–f, 6a–f.

The third configuration was employed for the $L = 22$ spin crystal ($N = 27$ ion chain) as depicted in Supplementary Fig. 8c. In this setup, we detuned $\Delta = -45 \times 2\pi$ kHz below the zig-zag mode, set $J = 0.2 \times 2\pi$ kHz and achieved relatively uniform nearest neighbor (NN) spin bonds by configuring the relative Rabi frequencies as $\Omega_j/\Omega_1 = \{1.0, 0.59, 0.59, 0.4, 0.47, 0.37, 0.50, 0.44, 0.60, 0.51, 0.68, 0.54, 0.68, 0.52, 0.62, 0.45, 0.53, 0.39, 0.49, 0.42, 0.61, 0.60 | 1 \le j \le 22\}$. The calculated $\bar{J}(d)$ is presented in Supplementary Fig. 8d (squares) and can be approximated by the curve $\bar{J}(d) = 2.7J \times e^{-|i-j|}$. This configuration was used in generating Fig. 2 and Supplementary Fig. 3. In this configuration, we apply the same staggered-phase correction to the array of individual optical beams as in the first configuration, to make the $J_{ij}$ coupling sign uniform.

Simultaneously with the Ising interaction, we apply the transverse magnetic field Hamiltonian[18,21]:

$$H_Z = \sum_{j=1}^{L} B_z^{(j)} \hat{s}_z^{(j)}. \tag{5}$$

We achieve independent control over the local magnetic field affecting each spin by shifting the optical frequency of the individually-addressing beam using its AOM channel. Since the spin is measured relative to the laser rotating frame, an instantaneous shift of the optical frequency of the $j$th beam by $B_z^{(j)}(t) \times 2\pi$ Hz relative to the carrier transition is equivalent to the application of such a transverse magnetic field in the spin frame. This shift introduces a small asymmetry in the detuning of the optical tones driving the red- and blue-sideband transitions, with $|B_z^{(j)}| \ll |\Delta|$. We shift and modulate the optical frequency to generate the magnetic fields that take the form:

$$B_z^{(j)}(t) = B_0 + \bar{\eta} \frac{z_0 \omega}{\sqrt{2}} \cos(\omega t) \cos(\varphi_j), \tag{6}$$

where $z_0 \approx 2.4$ is the first root of the 0th Bessel function of the first kind. We experimentally set $B_0 = 18J$ and $\omega = 6J$ for all configurations. We have verified that, in the absence of Raman beams, individual spins precess as expected under Eq. (6) by monitoring their precession in the $xy$ plane. Although not necessary in this experiment, our setup can also realize values $\bar{\eta} \ge 1$.

To accurately apply the magnetic field and eliminate unwanted nonuniform fields acting as an effective on-site potential, we calibrate and correct energy shifts, primarily from differential light shifts induced by the Raman beams. We characterize these shifts by tracking spin evolution when initialized along the $x$-axis and measured in the $x$-basis for $B_z^{(j)}(t) = 0$. From this, we determine the necessary offset fields to minimize undesired evolution and apply static corrections by adjusting the optical frequency of each individually addressed beam. We also ensure that the amplitudes of the red and blue tones are balanced, to minimize light shift noise in the presence of the Ising interaction.

We measure each $J_{ij}$ element in Fig. 1a–c by turning on the two beams addressing the $i$th and $j$th ions while turning off all other beams in the array. The ions are initialized in the state $|\uparrow_z^{(i)} \downarrow_z^{(j)}\rangle$ for $j > i$. Unlike ref. 21, here, the magnetic field in Eq. (6) is applied during the measurement, along with a constant-amplitude pulse, and oscillations in the populations are measured. We fit the average staggered magnetization $\langle \hat{s}_z^{(i)} - \hat{s}_z^{(j)} \rangle$ to the function $\exp(-\Gamma_{ij}t) \cos(\pi J_{ij}t)$ using $J_{ij}$ and $\Gamma_{ij}$ as fitting parameters.

## Dimerization of bonds by the Floquet drive

The transverse field Ising model under consideration is effectively described by the Hamiltonian in Eq. (1). This representation is achieved through a frame transformation that rotates each spin by its Larmor frequency corresponding to its local transverse field. Given that the applied transverse fields dominate the Ising interaction ($B_z^{(j)} \gg J$), we can express it as the Ising Hamiltonian using the raising and lowering spin operators $\hat{s}_x^{(i)} \hat{s}_x^{(j)} \approx \frac{1}{2}(\hat{s}_+^{(i)} \hat{s}_-^{(j)} + \hat{s}_-^{(i)} \hat{s}_+^{(j)})$. This transformation introduces fast oscillating terms involving $\hat{s}_\pm^{(i)} \hat{s}_\pm^{(j)}$[18]. We note that such terms can induce transitions outside the single-excitation subspace with the sublattice structure defined below, but their effects are strongly suppressed when $B_0 \gg J$. This is evident from comparison of the evolution of the static SSH spin Hamiltonian in Eq. (1) and the full time dependent Hamiltonian in Eq. (2) as shown in Supplementary Figs. 2, 7 and 9.

The periodic fields applied to the system modify the bare spin interaction matrix $J_{ij}$ and yield a scaled interaction matrix $\mathcal{J}_{ij}$[14].

$$\mathcal{J}_{ij} = j_0 \left( 2\eta \sin\left(\frac{\pi}{4}(i+j) + \phi\right) \sin\left(\frac{\pi}{4}(i-j)\right) \right) J_{ij}. \tag{7}$$

Here, $j_0(x)$ denotes the 0th Bessel function of the first kind and $\eta = \frac{z_0 \bar{\eta}}{\sqrt{2}}$. Equation (7) shows that the suppression acts as a multiplicative factor. To leading order, it depends only on the parameters of the Floquet field, such as its amplitude and phase. It is largely independent of other experimental factors, most notably, the phonon mode spectrum, which determines the bare interaction coefficients $J_{ij}$ (see Eq. (4)). We therefore attribute the imperfect suppression observed in some bonds (e.g., Fig. 1c) to small deviations in the applied Floquet field parameters, which can potentially be improved with more precise calibration.

We now discuss the sub-lattice structure of the model. We assume that $L$ is even and divide the chain into odd (A sub-lattice) and even sites (B sub-lattice), as illustrated in Supplementary Fig. 1. We express the $L \times L$ interaction matrix $\mathcal{J}_{ij}$ in terms of four $L/2 \times L/2$ sub-matrices, with elements denoted as $\mathcal{J}_{An;Am}$, $\mathcal{J}_{Bn;Bm}$, $\mathcal{J}_{An;Bm}$, $\mathcal{J}_{Bn;Am}$. Here $\mathcal{J}_{An;Bm}$ denotes the coupling between site $n$ in sublattice A, and site $m$ in sublattice B, and so on. At $\phi = \frac{\pi}{4}$ and $\phi = \frac{3\pi}{4}$, ref. 14 showed that the Zak phase of the long-range SSH model is quantized; in particular, at $\phi = \frac{3\pi}{4}$, the system supports a topologically non-trivial state. For these phase values, we obtain:

$$\mathcal{J}_{An;Am} = \mathcal{J}_{Bn;Bm} = \bar{J}(2d)D(d), \tag{8}$$

where $d = n - m$, and $D(d)$ is a dimerization parameter that takes the values $D(d) = j_0(0) = 1$ if $d$ is even, and $D(d) = j_0(\bar{\eta} z_0)$ if $d$ is odd. Equation (8) implies that the chain possesses inversion symmetry, assuming that the original non-dressed interaction is homogeneous (i.e., $J_{nm} = \bar{J}(d)$), a condition not fulfilled for values $\phi \neq \frac{\pi}{4}, \frac{3\pi}{4}$. We also have non-diagonal blocks that connect the two sub-lattices and represented by:

$$\mathcal{J}_{An;Bm} = \bar{J}(2d-1)\bar{D}(d), \tag{9}$$

with a different dimerization parameter $\bar{D}(d)$. If $d$ is even, it takes values $\bar{D}(d) = j_0(0)$ for $\phi = \frac{\pi}{4}$ and $\bar{D}(d) = j_0(\bar{\eta} z_0)$ for $\phi = \frac{3\pi}{4}$. If $d$ is odd, $\bar{D}(d) = j_0(\bar{\eta} z_0)$ for $\phi = \frac{\pi}{4}$ and $\bar{D}(d) = j_0(0)$ for $\phi = \frac{3\pi}{4}$.

The bulk properties of the system are described in the plane-wave basis with momentum $k$ in which the dressed interaction matrix can be written in terms of the four blocks:

$$\mathcal{J}(k) = \begin{pmatrix} E(k) & \Delta(k) \\ \Delta^*(k) & E(k) \end{pmatrix} \tag{10}$$
$$= E(k)\mathbb{I} + \Delta(k)\sigma^+ + \Delta^*(k)\sigma_-.$$

with $E(k) = \sum_d \bar{J}(2d)D(d)e^{ikd}$, and $\Delta(k) = \sum_d \bar{J}(2d-1)\bar{D}(d)e^{ikd}$. The matrix $\mathcal{J}(k)$ possesses both time-reversal and inversion symmetry,

$$\mathcal{J}(k) = \mathcal{J}(-k)^*,$$
$$\mathcal{J}(k) = I\mathcal{J}(-k)I, \tag{11}$$

with the inversion operator $I = \sigma_x$[42-44].

In the short-range limit ($\bar{J}(d) = 0$ beyond nearest-neighbors, $d > 1$), the matrix $\mathcal{J}_{ij}$, corresponds to the couplings of the standard SSH model with $\epsilon(k) = 0$ and $\Delta(k) = \bar{J}(1)(\bar{D}(0)e^{ikd} + \bar{D}(1)e^{-ikd})$. If we adiabatically deform the matrix $\mathcal{J}(k)$ by increasing the range of interactions, the number of edge states—which, in this model, depends on the Zak phase—will be conserved. Such adiabatic deformation of $\mathcal{J}(k)$ is valid as long as the gap remains open and the symmetries of the system are preserved, a condition guaranteed at $\phi = \frac{\pi}{4}, \frac{3\pi}{4}$.

To illustrate this scenario, we consider the eigenstates of the matrix $\mathcal{J}$ for an exponentially decaying interaction profile:

$$J_{ij} = Je^{-|i-j|/\xi} \tag{12}$$

with a non-zero dimerization parameter. In the limit $\xi \ll 1$, we are in the standard short-range SSH model (see Supplementary Fig. 10 for a crystal of $L = 50$ spins). Dimerization induces two zero-energy mid-gap states. By adiabatically deforming the interaction range of this Hamiltonian while keeping $\phi = \frac{3\pi}{4}$, we find that the two degenerate states survive. However, when the interaction is so long-range that the gap is too small, the conditions for adiabatic deformation are no longer met, and the edge-states merge with the bulk and disappear.

## Fermionic representation

Spin SSH models can be represented in terms of free fermions, such that their dynamics is governed by single-particle physics. Long-range terms lead to fermion-fermion interactions. This fermionic representation relies on the Jordan-Wigner transformation,

$$\hat{s}_-^{(j)} = \frac{1}{2}\hat{c}_j e^{-i\pi \sum_{j' < j} \hat{c}_{j'}^\dagger \hat{c}_{j'}},$$
$$\hat{s}_+^{(j)} = \frac{1}{2} e^{i\pi \sum_{j' < j} \hat{c}_{j'}^\dagger \hat{c}_{j'}} \hat{c}_j^\dagger, \tag{13}$$
$$\hat{s}_z^{(j)} = \hat{c}_j^\dagger \hat{c}_j - \frac{1}{2}.$$

Here $\hat{c}_j$ and $\hat{c}_j^\dagger$ are the fermionic annihilation and creation operators in site $j$, respectively. The long-range spin Hamiltonian in Eq. (1) casts as,

$$H = \sum_{i,j} \hat{c}_i^\dagger \hat{\mathcal{J}}_{ij} \hat{c}_j, \tag{14}$$

with the modified interaction matrix

$$\hat{\mathcal{J}}_{ij} = \mathcal{J}_{ij} \prod_{i < j' < j} \left( 2\hat{c}_{j'}^\dagger \hat{c}_{j'} - 1 \right). \tag{15}$$

For the scenario of short range interactions, the spin SSH Hamiltonian is equivalent to the free fermion one, since $\hat{\mathcal{J}}_{i,i+1} = \mathcal{J}_{i,i+1}$. However, for long range interaction, spin-spin interactions $\hat{\mathcal{J}}_{i,i+r}$ with $r > 1$ include string operators that depend on the fermionic number operators at intermediate sites. This leads to many-body fermion-fermion interaction terms, rendering the Hamiltonian in Eq. (14) non-quadratic in the fermionic operators. Our numerical simulation of the long-range, free-fermionic Hamiltonian, shown in Fig. 3d–f and Supplementary Fig. 4 (magenta curve) is realized by solving the evolution governed by the fermionic Hamiltonian in Eq. (14) for $\hat{\mathcal{J}}_{ij} = \mathcal{J}_{ij}$, thus neglecting the string operators that lead to fermion-fermion interactions.

## Domain wall dynamics

The domain wall evolution observed in Fig. 4 can be understood by expressing the interaction between and within the domains. The initial state of the spin crystal is represented as $|\psi_0\rangle = |\uparrow\rangle_1 \dots |\uparrow\rangle_{\frac{L}{2}-1} |\downarrow\rangle_{\frac{L}{2}} \dots |\downarrow\rangle_L \equiv |\uparrow\rangle_L |\downarrow\rangle_R$, and the interaction matrix is decomposed into $\mathcal{J} = \mathcal{J}_L + \mathcal{J}_R + \mathcal{J}_{LR}$. Here, $\mathcal{J}_L$ and $\mathcal{J}_R$ are the interaction matrix within the left-side ($j = 1, \dots, L/2$) and the right-side spins ($j = L/2 + 1, \dots, L$), respectively, and $\mathcal{J}_L$ is the interaction matrix between the left and right sides. The initial state is an eigenstate of the spin Hamiltonian corresponding to uncoupled halves, satisfying

$$\sum_{ij} \left(\mathcal{J}_L + \mathcal{J}_R\right)_{ij} \hat{s}_+^{(i)} \hat{s}_-^{(j)} |\psi_0\rangle = 0. \tag{16}$$

However, the term $\mathcal{J}_{LR}$ couples $|\psi_0\rangle$ to excited states $|\psi_n\rangle_L |\psi_m\rangle_R$ with energy $\epsilon_n + \epsilon_m$. These excited states are defined as spin waves $|\psi_n\rangle_L = 2\sum_{j\in L}\psi^L_{n,j}\hat{s}^{(j)}_-|\uparrow\rangle_L$ and $|\psi_n\rangle_R = 2\sum_{j\in R}\psi^R_{n,j}\hat{s}^{(j)}_+|\downarrow\rangle_R$, with energies given by $\mathcal{J}_L\psi^L_n = \epsilon_n\psi^L_n$ and $\mathcal{J}_R\psi^R_n = \epsilon_n\psi^R_n$. The coupling strength between the initial domain wall state and these excited spin-wave states is given by $2\sum_{ij}(\mathcal{J}_{LR})_{i\in L, j\in R}\psi^L_{n,i}\psi^R_{n,j}$. For the case of short-range interaction and strong dimerization, this coupling is stronger when $n$ and $m$ are edge-states on the boundary between each of the sides, explaining the coherent oscillation observed in Fig. 4b. When long-range terms are introduced, long-range couplings are activated between the initial state and spin-wave states $n$ and $m$ n the bulk, explaining the spread of excitations along both sides of the chain.

## Numerical simulations and analysis

We numerically simulate the Unitary dynamics of the $L = 12$ crystal corresponding to the full time-dependent Hamiltonian in Eq. (2) or the static Hamiltonian in Eq. (1) using the experimental parameters described in "Methods". We calculate the mode participation matrix elements by solving the Laplace equation assuming the trapping potential described above, from which we find the positions of the ions. Then through linearization we determine the mode participation matrix elements, used for the calculation of the $J_{ij}$ matrices. We represent the spin matrices, the Hamiltonian, and the quantum spin state using the sparse package in Matlab, and solve the Unitary evolution using standard ordinary differential equations (ODE45) solver.

To quantify the thermalization process due to interaction, we compute the thermalization rate as the inverse of the slope of the linear line connecting the regions with magnetization above and below the equilibrium value of $\langle s_z^{(j)}\rangle = \frac{1}{2L}$ for $L = 12$. We then use a line of $\tau = (j-1)/v_s$ to fit the boundary in the image, where $j$ is the ion index and $\tau = Jt/\pi$ is the evolution time. The fitted slope is presented as a yellow line in Fig. 1d–f and Supplementary Fig. 2a–c. Dashed lines show the values for the 95% confidence interval of the fit.

## Data availability

The data displayed in Figs. 1–4 are publicly accessible at this repository https://doi.org/10.57760/sciencedb.27829.

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

## Acknowledgements

We thank Marko Cetina, Victor Gurarie, Pedro Nevado and Alvaro Gomez for fruitful discussions. This work is supported by the DOE Quantum Systems Accelerator (DE-FOA-0002253), the NSF STAQ Program (PHY-1818914), the NSF QLCI Center on Robust Quantum Simulations (QISET-2120757), the AFOSR MURI on Dissipation Engineering in Open Quantum Systems (FA9550-19-1-0399), the Spanish project PID2021-127968NB-I00 funded by MCIN/AEI/10.13039/501100011033/FEDER, UE, and the CSIC Quantum Technologies Platform PTI-001.

## Author contributions

O.K., L.F., and C.M. contributed to the development of the experimental setup. O.K. and L.F. performed the experiments. O.K. and D.P. carried out the numerical simulations. L.F. and D.P. analyzed the experimental data. D.P. contributed to the theoretical analysis. C.M. supervised the research and secured funding. All authors contributed to the conception, design, interpretation, and writing of the manuscript.

## Competing interests

C.M. is a founder of IonQ, Inc. and has a personal financial interest in the company. All other authors declare no competing interests.
