## [Transparent Peer Review file · Nature Communications]

Floquet Control of Interactions and Edge States in a Programmable Quantum Simulator

Corresponding Author: Professor Or Katz

Version 0:

Reviewer comments:

Reviewer #1

(Remarks to the Author)

--- What are the noteworthy results? ---

In this paper, the authors were showing their observation of quantum dynamics in a trapped ion quantum simulator. In particular, the authors were able to apply site-specific Floquet fields that control both the dimerization and the range of the spin-spin interaction. The authors were following a proposal as proposed in reference 31, and studied the dynamics in this system. The authors were able to create localized edge states in a trapped ion system.

--- Will the work be of significance to the field and related fields? How does it compare to the established literature? If the work is not original, please provide relevant references. ---

This work is of significance in that the authors were able to have single-site addressing of trapped ion quantum simulators to study quantum dynamics of interacting quantum systems. The authors have also demonstrated their ability to use site-specific Floquet engineering to create localized edge states. This work potentially opens up interesting direction for trapped ion simulators.

--- Does the work support the conclusions and claims, or is additional evidence needed? ---

The work claims to explore the topological properties of SSH model with different types of interactions. However, while I am convinced that the authors have a system with alternating interactions and with tunable range of interactions, whether this system is topological is questionable.

1) The authors focused mainly on comparing the bulk and edge states dynamics, for example in Figure 1d-f. The authors then claimed that having different dynamics means the system is topological, and thus the edge states they observed is the topological edge state. However, there are 1D models that are not topological insulators, for example, the Rice-Mele model that breaks the chiral symmetry. The Rice-Mele model also has localized edge states that can exhibit the dynamics observed in Figure 1d-f and Figure 3ab. The authors showed in Figure 1h that the theory calculation and experimental observations agree. However, the theory calculation, as indicated by "NUMERICAL SIMULATIONS AND ANALYSIS" at the end of the paper, was to simulate the full time-dependent Hamiltonian, rather than the underlying SSH model, this, combined with the fact that several experimental data points in Figure 1h are somewhat far away from the theory calculation, raises questions: (1) if the theory calculation indeed gives an SSH model, does the discrepancy between theory and experiment indicates that the authors don't have an SSH model? And (2) if the discrepancy between theory and experiment is negligible, is the theory calculation indeed simulating an SSH model?

3) The authors were trying to show the symmetry of their system by showing symmetry of the spin bond. However, having an alternating value of J_{ij} does not guarantee that these interaction amplitudes have the same phase, which is needed for the SSH model. The authors also did not show/comment on whether the on-site potential of each spin is uniform — if not, this breaks the chiral symmetry and we also do not have an SSH model.

4) The discussions around interaction is also confusing. Why did the authors say that there is a "delicate balance" between an interaction (that does not break the chiral symmetry) and the topological nature of the Hamiltonian? I understand that the edge states get less robust, but how do the topological invariances, such as the Zak phase or the winding number, change?

5) The fast oscillation terms involving $s^{(i)}_{\pm} s^{(j)}_{\pm}$, do they break the chiral symmetry of the SSH model?

6) Moreover, the authors have made some confusing claims about topological insulators. For example:

- (1) One of the biggest issues is that the authors seem to think that when a 1D chain has chiral symmetry, it automatically has edge states. This is wrong. For the SSH model, the model itself has chiral symmetry, but it has topologically trivial (no edge states) and non-trivial (has edge states) phases.
- (2) "The edge shows greater protection at $\phi = \pi/4$ and $\phi = 3\pi/4$ where the Hamiltonian features a reflection symmetry leading to Zak phase quantization and topologically protected (quasi-) zero-energy modes." In addition to the fact that the figure does not support this claim, it is also wrong to say that "the Hamiltonian features a reflection symmetry" leads to "Zak phase quantization and topologically protected (quasi-) zero-energy modes". For one, a system with reflection symmetry does not guarantee topology. Secondly, the Zak phase is gauge-dependent.
- (3) "A key characteristic of topological insulators is the distinct response of excitations at the edges compared to those within the crystal's bulk," and "the enhanced persistence of spin excitations at the edges ($j = 1$ or $j = 22$), compared to the bulk of the crystal ($2 \leq j \leq 21$), serves as a distinctive hallmark of the crystal's topological state.": It is true that if a topological insulator is in its non-trivial phase, we see different edge/bulk properties. However, simply observing different edge/bulk properties does not mean we have observed something topological.
- (4) "These results underscore the significance of the site-dependent pattern of the Floquet field and the reflection symmetry that imparts topological properties to the Hamiltonian." The reflection symmetry does not impart topological properties of a 1D system.
- (5) Inversion symmetry in equation (8) is wrong.
- (6) In the method section, when discussing adiabatic deformation of $\mathcal{H}(k)$, the author claim that the number of edge states is a topological invariant, which is wrong. One can have two edge states in the Rice-Mele model, which does not have a quantized winding number.

7) Below are two small points that are confusing:

- (1) The authors mentioned: "we present another intriguing multiple-excitation scenario where the spin excitations are ordered to form two domain walls. The left half of the $L = 12$ crystal are oriented upwards and those in the right half point downwards." Where is the second domain wall?
- (2) The authors mentioned: "As we increase the amplitude of the Floquet drive ..." which figure?

--- Are there any flaws in the data analysis, interpretation and conclusions? ---

Yes.

1) Fig 2d, when $\phi = \pi/4$, their conclusion in the caption is wrong.

2) More justification needed to show that they have a topological insulator. Some analysis to justify a topological insulator could be: measure the winding number/Zak phase; show topological phase transition; show edge states are indeed zero-energy and the bulk is an insulator; show energy gap closing at topological phase transition, etc.

--- Do these prohibit publication or require revision? ---

Yes. Major revision required.

--- Is the methodology sound? Does the work meet the expected standards in your field? ---

I believe the authors are experimentally simulating the time-dependent hamiltonian they mentioned in the "numerical simulation" part. Whether it is a topological insulator is unclear.

--- Is there enough detail provided in the methods for the work to be reproduced? ---

Perhaps. I do wonder if the theoretical simulation has any approximations. And, if the theoretical simulation discussed at the end of the paper indeed simulates an SSH model, namely, is the effective Hamiltonian an SSH model. Or, how does a theoretical calculation using SSH model (i.e. time-independent models) compare to the experimental observations?

Reviewer #2

(Remarks to the Author)

The manuscript by Katz et al. presents an experimental exploration of topological insulator phases based on a modified SSH model. They make use of their programmable trapped-ion simulator/computer in which where they implement the transverse-field Ising Hamiltonian. A key enabling part is their single-ion addressing system which allows for simultaneous illumination of all ions and for phase and amplitude control of the transverse field for each ion individually. By means of Floquet engineering they can render the XY Hamiltonian reflection-symmetric in order to observe topologically-protected edge states. In spin chains with up to 22 ions, they investigate properties of these edge states, also with initial states with multiple excitations such as a Neel state or a state consisting of two domain walls.

In my opinion, the results are well presented and sufficiently novel to justify a publication in Nature Communications. In particular the multi-excitation scenarios are a valuable extension to previous studies which show the great capabilities of their experimental platform with the site-specific control of the spin-spin interactions. However, before I finally recommend to publish the manuscript, a few issues should be addressed by the authors.

In the main text:

(major) The introduction is rather compact and requires some expert knowledge about topological insulators. Since Nature Communications is a journal for a broader audience, I would suggest to make the topic more accessible for this audience (e.g. present the underlying Hamiltonian of the modified SSH model as well as its properties and eigenstates, further

comments on which properties of this model have been explored so far in quantum simulation and what is shown here on top, etc.)

(major) In Fig 1c the coupling strengths of the “undesired” bonds seems to be highly suppressed but still existing. Is this due to coupling to motional modes other than the zig-zag mode or is there another (technical) reason? How much does this effect influence the excitation spreading rate?

(minor) In Fig. 1d-f I find the different shapes of the orange and blue lines a bit confusing. It makes the impression as if there would be a fundamental difference between them, however, in my understanding they merely indicate the NN and NNN interaction strength. I would suggest to show both with a similar shape.

(minor) In the caption of Fig 1. you label $\hbar * J$ as the spin bond energy, on the figure axes it is only J.

(major) I have some difficulties in understanding the data shown in Fig 2c. If the grey dash line is the late-time averaged spin excitation of the whole crystal, why does it vary with the ion number index? Does each point exclude the ion with index j, i.e. the yellow boxes exclude the orange one? If yes, why is the grey dash line always lower than every data point calculated from the orange boxes? Furthermore, can you comment on the imperfections in your experiment which seem to give rise to a variance of the late-time averaged spin excitations of each ion which is much bigger than the error bars of the data points? And similar, which imperfections result in a spin excitation of the edge ions unequal from zero?

(minor) The data in Fig. 2 are measured with 22 ions and an η_{bar} of 0.8. Is there any technical reason to not go to $\eta_{\text{bar}} = 1.0$?

(minor) In Fig. 3 d-f the time evolution of three exemplary spins is shown. I don't see any big benefit in presenting ion 2 and ion 4 since their dynamics are pretty similar. I would suggest to show one of the edge spins instead to see how well the agreement between experiment and theory is in this case.

(major) The outlook is a bit too short and unspecific in my opinion. The authors mention some potential follow-up studies, however, these experiments could be supported e.g. by theory proposals or concrete ideas (e.g., “where exactly are the benefits compared to atomic or photonic experiments”, “which many-body ground-states could be investigated”, etc.) Moreover, questions such as “Which current limitations of the experiment hinder these follow-up studies” or “What are the experimental steps and modifications required to realize them” might be interesting to answer.
In the appendix Methods part:

(minor) In the first paragraph the authors mention the two laser tones being nearly symmetric to the motional modes. I find “nearly” a bit vague. Is it symmetric to the sidebands or not? If not, is there any reason (e.g., deliberate detuning for Stark shift compensation, etc.)?

(major) In the same section they describe the use of auxiliary ions to ensure more equal spacing between the ions. Is there any reason why they use an odd number of ions for an even number of spins with the result of different numbers of auxiliary ions on the two ends of the chain?

(minor) There are at least 5 typos in the main text and appendix, please revise the manuscript.

(minor) It would be good to clearly distinguish between frequencies and angular frequencies. For example, in the ‘additional experimental details’ section, the com mode frequency is given as an angular frequency ($\omega_1 = (2\pi) 3.08 \text{ MHz}$), but then the detuning Δ is written as $\Delta = 99 \text{ kHz}$. Given eq. 2, I assume that Δ is supposed to be an angular frequency, too, which would be realized by detuning the laser by about 16 kHz from the sideband. Why not write all angular frequencies in the manuscript with an explicit factor of 2π ($\Delta = (2\pi) * \dots$) in order to avoid such ambiguities?

Reviewer #3

(Remarks to the Author)

Version 1:

Reviewer comments:

Reviewer #1

(Remarks to the Author)

I thank the authors' efforts to address my questions and comments. I now recommend the publication of this manuscript.

Reviewer #2

(Remarks to the Author)

The revised version of the manuscript and the author reply address the points we raised in our report in a satisfactory manner. Some of the bolder claims made in the first version of the manuscript were softened by the authors in response to criticism from the other referee. Nevertheless, we believe that the current manuscript still contains results that are sufficiently novel to justify its publication in Nature Communications.

Reviewer #1:

--- What are the noteworthy results? ---

In this paper, the authors were showing their observation of quantum dynamics in a trapped ion quantum simulator. In particular, the authors were able to apply site-specific Floquet fields that control both the dimerization and the range of the spin-spin interaction. The authors were following a proposal as proposed in reference 31, and studied the dynamics in this system. The authors were able to create localized edge states in a trapped ion system.

--- Will the work be of significance to the field and related fields? How does it compare to the established literature? If the work is not original, please provide relevant references. ---

This work is of significance in that the authors were able to have single-site addressing of trapped ion quantum simulators to study quantum dynamics of interacting quantum systems. The authors have also demonstrated their ability to use site-specific Floquet engineering to create localized edge states. This work potentially opens up interesting directions for trapped ion simulators.

We thank the Referee for highlighting the significance of our work in studying the dynamics of interacting spin models with site-specific and time-dependent control with Floquet engineering.

--- Does the work support the conclusions and claims, or is additional evidence needed? ---

The work claims to explore the topological properties of SSH model with different types of interactions. However, while I am convinced that the authors have a system with alternating interactions and with tunable range of interactions, whether this system is topological is questionable.

This is an important point. The theoretical study in Ref. [Nevado et al. Phys. Rev. Lett. 119, 210401 (2017)] (previously Ref. 31) demonstrates that the static Hamiltonian in Eq. 1 can host topological phases, as indicated by a nontrivial Zak phase in certain parameter regimes. However, while our experimental results are consistent with and indicative of expected topological properties, they do not strictly prove that the system is in a topological phase of matter. Instead, our results demonstrate the dynamics of a spin-based variant of the SSH model, including edge-state behavior.

To eliminate ambiguity or any controversy regarding measurement of topological properties, we have substantially revised the manuscript to clarify the interpretation of our experimental observations. We now explicitly highlight that we observe Floquet-induced dimerization of the bonds and edge-state dynamics, but do not directly measure topological invariants. These revisions include extensive updates to the title, abstract, and main text. Below, we address the specific technical points raised by the Referee.

1) The authors focused mainly on comparing the bulk and edge states dynamics, for example in Figure 1d-f. The authors then claimed that having different dynamics means the system is topological, and thus the edge states they observed is the topological edge state. However, there are 1D models that are not topological insulators, for example, the Rice-Mele model that breaks the chiral symmetry. The Rice-Mele

model also has localized edge states that can exhibit the dynamics observed in Figure 1d-f and Figure 3ab.

We agree with the Referee that the presence of edge-state dynamics alone does not strictly indicate topological properties, and we have revised the manuscript to clarify this distinction. However, for the static model in Eq. 1, the analysis in Ref. [Nevado et al. Phys. Rev. Lett. 119, 210401 (2017)] suggests that, in the absence of imperfections, edge states and a topological phase can emerge under specific conditions, including chiral symmetry (for short-range interactions) or reflection symmetry (for long-range interactions), and particular bond structures (e.g., $\phi = 3\pi/4$). We now directly emphasize that while our measurements do not establish that the observed edge states originate from a topological phase, they are consistent with the expected properties of the model we study. For example in the Caption of Fig. 2 we now state: *“The generalized SSH model in Ref. [Nevado et al. Phys. Rev. Lett. 119, 210401 (2017)] predicts a nontrivial topological phase at this value. While we do not directly measure a topological invariant, our observations are consistent with edge-state behavior in such a phase.”*

Nonetheless, we believe our results are significant regardless of whether they are classified as topological, as they experimentally demonstrate how precision Floquet engineering enables control over bond dimerization, creating an interaction structure that, despite experimental imperfections, approximately retains the required symmetries and supports edge-state localization and dynamics. The combination of this level of control with Floquet engineering opens new possibilities for quantum simulations of interacting spin systems.

The authors showed in Figure 1h that the theory calculation and experimental observations agree. However, the theory calculation, as indicated by “NUMERICAL SIMULATIONS AND ANALYSIS” at the end of the paper, was to simulate the full time-dependent Hamiltonian, rather than the underlying SSH model, this, combined with the fact that several experimental data points in Figure 1h are somewhat far away from the theory calculation, raises questions: (1) if the theory calculation indeed gives an SSH model, does the discrepancy between theory and experiment indicates that the authors don’t have an SSH model? And (2) if the discrepancy between theory and experiment is negligible, is the theory calculation indeed simulating an SSH model?

We address this point by extending our numerical analysis to include simulations of the dynamics in Fig. 1 based on the static SSH Hamiltonian in Eq. 1 in addition to the time-dependent Hamiltonian (now Eq. 2). The new simulation results, now presented in Extended Data Fig. 2, allow direct comparison with the full time-dependent Hamiltonian for the experimental configuration.

We find that the static SSH Hamiltonian closely reproduces the dynamics observed in the full time-dependent model, indicating that the deviations observed in Fig. 1h are primarily due to experimental imperfections rather than a fundamental departure from an SSH-like Hamiltonian. This corresponds to case (1) as suggested by the Referee.

The observed discrepancy likely arises from fluctuations in the power of the laser field responsible for generating the spin-spin interaction across different experimental runs. Since this laser power is not

actively regulated, such fluctuations can lead to global rescaling of the J matrix, effectively altering the timescale of the evolution. This would modify the observed speeds in Fig. 1h and only to a lesser extent affect the underlying Hamiltonian structure. We believe that implementing laser power stabilization and further improving experimental hardware will reduce these deviations in future experiments. We now discuss these results and experimental limitations in the revised manuscript.

3) The authors were trying to show the symmetry of their system by showing symmetry of the spin bond. However, having an alternating value of $|J_{ij}|$ does not guarantee that these interaction amplitudes have the same phase, which is needed for the SSH model.

This is indeed an important comment that deserves emphasis. The interaction Hamiltonian we experimentally generate follows the standard long-range Ising model and is given by:

$H_{XX} = 2 \sum_{ij} J_{ij} s_x^{(i)} s_x^{(j)}$, where the coupling coefficients J_{ij} in Eq. 2 are real-valued. This formulation follows the approach presented in Ref. [Monroe et al., Rev. Mod. Phys. 93 025001 (2021)] where $\Omega_i = |\Omega_i|$ represents real-valued, positive rabi amplitudes. Importantly, any phase information associated with the laser fields does not appear in J_{ij} , but rather affects the spin and phonon operators (see e.g., Eq. 7 in [Monroe et al., Rev. Mod. Phys. 93 025001 (2021)]). Consequently, J_{ij} does not carry a complex phase factor and can only be positive or negative.

We now clarify the conditions under which the J_{ij} matrix effectively behaves with a uniform phase factor across the three experimental configurations considered in this work (see Extended Data Fig. 5).

For long-range interactions (Configuration 2, Extended Data Fig. 5b), the interaction is engineered by tuning the Raman detuning to couple predominantly to long-wavelength phonon modes [near the center-of-mass (COM) mode], leading to all J_{ij} elements having the same sign. This naturally results in a uniform phase factor across the system.

In Configurations 1 and 3 for short-range interactions, however, the sign of J_{ij} alternates and follows the pattern $J_{ij} = (-1)^{i-j-1} |J_{ij}|$. This staggered structure, evident in Extended Data Fig. 5a and 5c, originates from coupling predominantly to short-wavelength phonon modes (near the zig-zag mode). While this alternation might appear to introduce phase variations, it can be effectively made uniform through a simple transformation: we shift the optical phase of the odd-site individual beams by π (a static shift), which is equivalent to flipping the x-component of every odd-indexed spin ($s_x^{(i)} \rightarrow -s_x^{(i)}$ for odd i), as discussed also in Ref. [Schuckert et al. Nat. Phys. 21 374 (2025)]. Mathematically, this corresponds to a staggered $R_z^{(i)}(\pi)$ rotation (i.e. acting only on odd i). This transformation removes the sign alternation in J_{ij} while keeping all other properties of the system unchanged (e.g., the field terms in the Hamiltonian), since $R_z(\pi)$ commutes with $s_z^{(i)}$.

In the revised manuscript, we now explain in detail in the Methods section that the evolution is realized in phase-uniform J_{ij} and provide further detail for the different configurations.

The authors also did not show/comment on whether the on-site potential of each spin is uniform — if not, this breaks the chiral symmetry and we also do not have an SSH model.

In our experiment, the effective magnetic field along the z-direction acts as an effective on-site potential for the spin degrees of freedom. We ensure its uniformity by performing a dedicated calibration procedure before data collection, actively compensating for any static sources of nonuniformity. Additionally, we verified that in the absence of the Ising interaction, the spins evolve as expected under the applied time-dependent $B_z^{(i)}(t)$ fields.

Our calibration procedure involves initializing the spins perpendicular to the effective magnetic field (e.g., along the x-axis) and measuring their evolution over time while the Ising interaction is turned on. If the on-site potential were nonuniform, the spins would exhibit uncontrolled precession that tilts them away from the x axis, which we explicitly monitor (by measuring the spin in the x-basis). The dominant source of such nonuniformity, in the absence of correction, arises from differential light shifts between the spin levels induced by the Raman beams. To mitigate this, we fine-tune the local effective magnetic field at each site to compensate for these light shifts, ensuring a uniform effective field across the chain. This site-specific control is a unique capability of our setup, relying on independent frequency control of each beam in the array. We now describe this calibration and compensation in the Methods section.

4) The discussions around interaction is also confusing. Why did the authors say that there is a “delicate balance” between an interaction (that does not break the chiral symmetry) and the topological nature of the Hamiltonian? I understand that the edge states get less robust, but how do the topological invariances, such as the Zak phase or the winding number, change?

The Zak phase or the winding number are typically defined on single-particle Hamiltonians. Thus, it can be defined within the single-spin excitation sector, or in the limit of free fermions (which is only valid in the case of short-range interactions). In the presence of interactions it is not immediately clear whether such localization will survive, since interactions can lead to couplings between the single particle localized states and the bulk. Therefore, our discussion in the methods section and Extended Data Fig. 7 focuses on how the single-particle phenomenology survives in the presence of interaction terms. We have revised the manuscript to better clarify this point.

5) The fast oscillation terms involving $s^{(i)} \otimes s^{(j)}$, do they break the chiral symmetry of the SSH model?

The fast-oscillating terms create pairs of spin excitations and drive the system out of the single-excitation subspace. As a result, extending the usual definition of chiral symmetry to the full Hamiltonian, including these terms, is not straightforward. In the SSH model, chiral symmetry is defined in terms of projectors into the two subspaces corresponding to the two sublattices. The chiral symmetry is fulfilled because the

SSH Hamiltonian only couples states in different sublattices. This definition relies on a bipartition of the Hilbert space only valid within the single-excitation subspace. Since the fast-oscillating terms couple spin states outside the single excitation subspace, we can conclude that they break the chiral symmetry by extending the Hilbert space beyond the subspace where this symmetry is appropriately defined.

Nonetheless, under the experimental parameters used in our setup, the influence of these terms is strongly suppressed. Their effect can be further reduced by increasing the strength of the uniform magnetic field. In the revised manuscript, we now note that these terms violate the symmetry but remain negligible due to suppression by the large field.

6) Moreover, the authors have made some confusing claims about topological insulators. For example:

(1) One of the biggest issues is that the authors seem to think that when a 1D chain has chiral symmetry, it automatically has edge states. This is wrong. For the SSH model, the model itself has chiral symmetry, but it has topologically trivial (no edge states) and non-trivial (has edge states) phases.

We completely agree with the referee and apologize for any confusion in the previous manuscript. The revised manuscript explicitly clarifies that chiral symmetry does not imply edge state.

(2) “The edge shows greater protection at $\phi=\pi/4$ and $\phi = 3\pi/4$ where the Hamiltonian features a reflection symmetry leading to Zak phase quantization and topologically protected (quasi-) zero-energy modes.” In addition to the fact that the figure does not support this claim, it is also wrong to say that “the Hamiltonian features a reflection symmetry” leads to “Zak phase quantization and topologically protected (quasi-) zero-energy modes”. For one, a system with reflection symmetry does not guarantee topology. Secondly, the Zak phase is gauge-dependent.

We thank the referee for this comment and have removed this claim to avoid any confusion. The new claim now makes a distinction between the experimental observation and the theory reference:

“The generalized SSH model in Ref. [Nevado et al. Phys. Rev. Lett. 119, 210401 (2017)] predicts a nontrivial topological phase at this value. While we do not directly measure a topological invariant, our observations are consistent with edge-state behavior in such a phase.”

(3) “A key characteristic of topological insulators is the distinct response of excitations at the edges compared to those within the crystal’s bulk,” and “the enhanced persistence of spin excitations at the edges ($j = 1$ or $j = 22$), compared to the bulk of the crystal ($2 \leq j \leq 21$), serves as a distinctive hallmark of the crystal’s topological state.”: It is true that if a topological insulator is in its non-trivial phase, we see different edge/bulk properties. However, simply observing different edge/bulk properties does not mean we have observed something topological.

We now discuss the comparison of the edge and bulk dynamics with no mention of topological properties. The first sentence was changed to: “We now turn to compare the response of excitations at the edges with those in the crystal’s bulk”. The second sentence was changed to “The observed

dynamics indicate enhanced persistence of spin excitations at the edges ($j=1$ or $j=22$) compared to the bulk ($2 \leq j \leq 21$)."

(4) "These results underscore the significance of the site-dependent pattern of the Floquet field and the reflection symmetry that imparts topological properties to the Hamiltonian." The reflection symmetry does not impart topological properties of a 1D system.

We have removed the claim about observation of topological properties from the paper.

(5) Inversion symmetry in equation (8) is wrong.

We thank the referee for spotting this error (the definition of inversion symmetry contained a wrong complex-conjugate symbol), and we have corrected Eq. 8 in the revised manuscript.

(6) In the method section, when discussing adiabatic deformation of $\mathcal{J}(k)$, the author claim that the number of edge states is a topological invariant, which is wrong. One can have two edge states in the Rice-Mele model, which does not have a quantized winding number.

We thank the referee for this comment and revised this claim. It now reads: "*If we adiabatically deform the matrix $\mathcal{J}(k)$ by increasing the range of interactions, the number of edge states - which, in this model, depends on the Zak phase - will be conserved*".

7) Below are two small points that are confusing:

(1) The authors mentioned: "we present another intriguing multiple-excitation scenario where the spin excitations are ordered to form two domain walls. The left half of the $L = 12$ crystal are oriented upwards and those in the right half point downwards." Where is the second domain wall?

We thank the Referee for pointing this out. We have corrected the text to read: "*we present another intriguing multiple-excitation scenario where the spin excitations are ordered to form two domains, separated by a single domain wall.*"

(2) The authors mentioned: "As we increase the amplitude of the Floquet drive ..." which figure?

We now indicated it is Figure 4.

--- **Are there any flaws in the data analysis, interpretation and conclusions?** ---Yes.

1) Fig 2d, when $\phi = \pi/4$, their conclusion in the caption is wrong.

We have removed this claim from the caption.

2) More justification needed to show that they have a topological insulator. Some analysis to justify a topological insulator could be: measure the winding number/Zak phase; show topological phase

transition; show edge states are indeed zero-energy and the bulk is an insulator; show energy gap closing at topological phase transition, etc.

The revised manuscript does not claim measurement of topological property but focuses on the experimental study of spin dynamics under controlled Floquet Hamiltonian. We discuss these potential measurements in the discussion.

--- Do these prohibit publication or require revision? ---

Yes. Major revision required.

We thank the Referee for the opportunity to submit a revised manuscript. We believe the current version better highlights the manuscript's contributions, particularly our ability to observe edge-state dynamics and more complex many-body states for dimerized bond patterns. Our experiment demonstrates exquisite temporal control over local fields at each spin site, a capability unique among quantum simulators. This level of control could potentially enable the study of more complex spin models in future studies.

--- Is the methodology sound? Does the work meet the expected standards in your field? ---

I believe the authors are experimentally simulating the time-dependent hamiltonian they mentioned in the "numerical simulation" part. Whether it is a topological insulator is unclear.

We have addressed this concern in the above statements concerning the classification of our system as relating to topological degrees of freedom. We appreciate this referee's comments on this issue.

--- Is there enough detail provided in the methods for the work to be reproduced? ---

Perhaps. I do wonder if the theoretical simulation has any approximations. And, if the theoretical simulation discussed at the end of the paper indeed simulates an SSH model, namely, is the effective Hamiltonian an SSH model. Or, how does a theoretical calculation using SSH model (i.e. time-independent models) compare to the experimental observations?

In the revision, we now present numerical results comparing both the time-dependent Hamiltonian (as implemented in the experiment) and the corresponding static Hamiltonian associated with the spin-version of the SSH model in Extended Data Fig. 2, in Supplementary Fig. 2 and Supplementary Fig. 3. The evolution of these two models for the set of parameters we used in the experiment is nearly identical within our experimental resolution, indicating that the system dynamics are well captured by the effective static Hamiltonian. This supports the interpretation that our experiment realizes the intended interaction structure.

Reviewer #2 & #3:

The manuscript by Katz et al. presents an experimental exploration of topological insulator phases based on a modified SSH model. They make use of their programmable trapped-ion simulator/computer in which where they implement the transverse-field Ising Hamiltonian. A key enabling part is their single-ion addressing system which allows for simultaneous illumination of all ions and for phase and amplitude control of the transverse field for each ion individually. By means of Floquet engineering they can render the XY Hamiltonian reflection-symmetric in order to observe topologically-protected edge states. In spin chains with up to 22 ions, they investigate properties of these edge states, also with initial states with multiple excitations such as a Neel state or a state consisting of two domain walls.

In my opinion, the results are well presented and sufficiently novel to justify a publication in Nature Communications. In particular the multi-excitation scenarios are a valuable extension to previous studies which show the great capabilities of their experimental platform with the site-specific control of the spin-spin interactions.

We thank the Referees for their thoughtful review and for highlighting the novelty of our multi-excitation results and the capabilities of our experimental platform.

However, before I finally recommend to publish the manuscript, a few issues should be addressed by the authors. In the main text:

(major) The introduction is rather compact and requires some expert knowledge about topological insulators. Since Nature Communications is a journal for a broader audience, I would suggest to make the topic more accessible for this audience (e.g. present the underlying Hamiltonian of the modified SSH model as well as its properties and eigenstates, further comments on which properties of this model have been explored so far in quantum simulation and what is shown here on top, etc.)

We have improved the manuscript's accessibility, as appropriately suggested. We have revised the introduction and added a section that presents the spin-based SSH Hamiltonian, highlighting its underlying physics. We have also revised the chain illustration (Extended Data Fig. 1) to visually show the dimerization pattern we consider.

(major) In Fig 1c the coupling strengths of the “undesired” bonds seems to be highly suppressed but still existing. Is this due to coupling to motional modes other than the zig-zag mode or is there another (technical) reason? How much does this effect influence the excitation spreading rate?

The suppression of undesired bonds primarily results from the properties of the applied Floquet field. Small imperfections in the local phase or amplitude of this field can lead to deviations from ideal suppression, and we attribute the residual couplings to such imperfections. We expect that these can be further reduced through more precise calibration of the Floquet parameters.

Coupling to motional modes beyond the targeted ones leads to non-uniformities in the bare interaction matrix J_{ij} prior to the application of the Floquet drive (as shown in Fig. 1a). After completing the data

collection, we developed an optimization technique that better accounts for the contributions of all motional modes by fine-tuning the local Raman beam amplitudes. This method, described in [De, et al. [arXiv:2410.13815 \(2024\)](https://arxiv.org/abs/2410.13815)], allows for more uniform spin-spin interactions and could improve future implementations.

We estimate that the influence of these additional motional modes on the bond suppression is limited, as the Floquet field acts as a multiplicative modulation applied to the J_{ij} matrix. In the revised manuscript, we clarify that the phonon mode spectrum sets the structure of the bare J_{ij} matrix, while the Floquet field defines site-dependent suppression factors, which together determine the effective J_{ij} .

(minor) In Fig. 1d-f I find the different shapes of the orange and blue lines a bit confusing. It makes the impression as if there would be a fundamental difference between them, however, in my understanding they merely indicate the NN and NNN interaction strength. I would suggest to show both with a similar shape.

Corrected.

(minor) In the caption of Fig 1. you label $\hbar J$ as the spin bond energy, on the figure axes it is only J .

Corrected: \hbar was removed from the caption.

(major) I have some difficulties in understanding the data shown in Fig 2c. If the grey dash line is the late-time averaged spin excitation of the whole crystal, why does it vary with the ion number index? Does each point exclude the ion with index j , i.e. the yellow boxes exclude the orange one? If yes, why is the grey dash line always lower than every data point calculated from the orange boxes?

We thank the Referees for pointing out this ambiguity. The grey line in Fig. 2c represents the late-time averaged spin excitation of the crystal, excluding the initially excited spin at site j (i.e., the yellow boxes exclude the orange one). This procedure leads to the variation of the grey dashed line as a function of initial excitation site index.

The difference between the black point (the late-time spin excitation at the initially excited site j) and the grey line (bulk average excluding site j) arises primarily from the dimerization of the spin-spin interactions and, from decoherence effects (which we explicitly address in the following comment). The strong dimerization (due to $\bar{\eta} = 0.8$) suppresses excitation transport, leading to partial localization of the initial excitation, which explains the persistent difference between the black and grey points for all values of j . While this effect is most pronounced at the edges, it remains visible throughout the bulk.

We have revised the manuscript text and caption of Fig. 2 to clarify these points.

Furthermore, can you comment on the imperfections in your experiment which seem to give rise to a variance of the late-time averaged spin excitations of each ion which is much bigger than the error bars

of the data points? And similar, which imperfections result in a spin excitation of the edge ions unequal from zero?

Our experiment has several imperfections that lead to deviation from the SSH model. First, the Ising interaction relies on the coupling to phonon modes. While operating in the off-resonant regime, the spin-phonon coupling is nonzero, manifesting as bit flip noise. This noise does not conserve the number of excitations, and is the leading decoherence observed in Fig. 2 and affecting the edge spins ($j=1$, and $j=22$). We also have noise on the strength of the coupling matrix J , resulting from fluctuations of the laser intensity and partially from ion heating interacting with tightly focused beams (see [Cetina, et al. PRX Quantum 3 010334 (2022)]) which we believe functions as the main source of error in Fig. 1h.

The error bars only account for statistical error due to a finite number of points. We now describe these experimental imperfections in the Methods section, and discuss mechanisms which can suppress them in future experiments.

(minor) The data in Fig. 2 are measured with 22 ions and an $\bar{\eta}$ of 0.8. Is there any technical reason to not go to $\bar{\eta} = 1.0$?

Technically, we can realize any value of $\bar{\eta}$ including 1. We now highlight this in the manuscript.

While planning the experiment we chose to present the data for $\bar{\eta} = 0.8$ because for $\bar{\eta} = 1$ the bonds are extremely localized throughout the crystal: in the edge they simply remain on the edge but in the bulk the excitation ideally oscillates with the nearest site and does not spread through the bulk. This renders the late time averaged less conclusive (As it depends what part of the oscillation is averaged), and the thermalization is not immediately evident. With $\bar{\eta} = 0.8$ there is a spread of excitation throughout the chain that we thought highlights better the thermalization process and the difference between edge and bulk dynamics.

(minor) In Fig. 3 d-f the time evolution of three exemplary spins is shown. I don't see any big benefit in presenting ion 2 and ion 4 since their dynamics are pretty similar. I would suggest to show one of the edge spins instead to see how well the agreement between experiment and theory is in this case.

We updated the figure to show the evolution of spin number 1 instead of 2. Additionally, we now show the evolution of all 12 spins in comparison to these models in Supplementary Figure 1.

(major) The outlook is a bit too short and unspecific in my opinion. The authors mention some potential follow-up studies, however, these experiments could be supported e.g. by theory proposals or concrete ideas (e.g., "where exactly are the benefits compared to atomic or photonic experiments", "which many-body ground-states could be investigated", etc.) Moreover, questions such as "Which current limitations of the experiment hinder these follow-up studies" or "What are the experimental steps and modifications required to realize them" might be interesting to answer.

We have appropriately expanded and revised the discussion section to include specific directions for future studies, including potential applications to quasi-crystalline spin models, Heisenberg-like interactions via time-dependent Floquet fields, and further study of topological properties such as Zak phase measurements and ground-state preparation. We also now discuss current experimental limitations and how improvements in beam control could enable studies with larger ion chains and more complex interaction patterns.

In the appendix Methods part:

(minor) In the first paragraph the authors mention the two laser tones being nearly symmetric to the motional modes. I find “nearly” a bit vague. Is it symmetric to the sidebands or not? If not, is there any reason (e.g., deliberate detuning for Stark shift compensation, etc.)?

The asymmetry between the red and blue frequency components of the Raman beams corresponds to the applied magnetic field amplitude driving the ions. The term “nearly” was intended to indicate that this asymmetry is small ($|B_z^{(j)}| \ll |\Delta|$).

To improve clarity, we have revised the Methods section to remove the term “nearly” and now state: *“When the detunings from the radial collective motional modes are symmetric, this configuration realizes the Ising Hamiltonian.”*

We also explicitly discuss the small asymmetry introduced by the transverse field in the revised text: *“This shift introduces a small asymmetry in the detuning of the optical tones driving the red- and blue-sideband transitions, with $|B_z^{(j)}| \ll |\Delta|$.”*

(major) In the same section they describe the use of auxiliary ions to ensure more equal spacing between the ions. Is there any reason why they use an odd number of ions for an even number of spins with the result of different numbers of auxiliary ions on the two ends of the chain?

This is a good point, with a subtle (highly experimental) answer. We use an even number of spins (ions that couple to Raman beams) to ensure that the two edges remain symmetric in terms of the dimerization pattern and to align better with the sublattice definitions. In principle, this condition can be satisfied with either an odd or even total number of ions. However, changing the total number of ions requires additional technical adjustments, such as fine-tuning the trapping potential to optimize alignment with the beams and re-optimizing sideband cooling for different motional mode configurations. Since we have already performed these technical optimizations in previous works (e.g., [Feng et al. Nature 623, 713 (2023)]), we chose to maintain the same number of ions to minimize technical overhead. This allowed us to focus on implementing and characterizing the time-dependent Floquet fields, which are the central novelty of this experiment.

In the revised manuscript, we now clarify that the experiment could be realized with either an odd or even number of auxiliary ions and that their use could be avoided if the individual beam spacing could be made non-uniform (e.g., through beam steering).

(minor) There are at least 5 typos in the main text and appendix, please revise the manuscript.

We have carefully revised the manuscript and corrected all identified typos and language inconsistencies.

(minor) It would be good to clearly distinguish between frequencies and angular frequencies. For example, in the 'additional experimental details' section, the com mode frequency is given as an angular frequency ($\omega_1 = 2\pi \cdot 3.08 \text{ MHz}$), but then the detuning Δ is written as $\Delta = 99 \text{ kHz}$. Given eq. 2, I assume that Δ is supposed to be an angular frequency, too, which would be realized by detuning the laser by about 16 kHz from the sideband. Why not write all angular frequencies in the manuscript with an explicit factor of 2π ($\Delta = (2\pi) \cdot \dots$) in order to avoid such ambiguities?

Done. The 2π factors are now explicitly included.

We thank again both referees for their review and comments.